# Nanoscale synchrotron x-ray analysis of intranuclear iron in melanised neurons of Parkinson's substantia nigra
Jake Brooks [1]✉, James Everett [1,2], Emily Hill [3], Kharmen Billimoria[1], Christopher M. Morris[4], Peter J. Sadler [5], Neil Telling [2] & Joanna F. Collingwood [1]

Neuromelanin-pigmented neurons of the substantia nigra are selectively lost during the progression of Parkinson's disease. These neurons accumulate iron in the disease state, and iron-mediated neuron damage is implicated in cell death. Animal models of Parkinson's have evidenced iron loading inside the nucleoli of nigral neurons, however the nature of intranuclear iron deposition in the melanised neurons of the human substantia nigra is not understood. Here, scanning transmission x-ray microscopy (STXM) is used to probe iron foci in relation to the surrounding ultrastructure in melanised neurons of human substantia nigra from a confirmed Parkinson's case. In addition to the expected neuromelanin-bound iron, iron deposits are also associated with the edge of the cell nucleolus. Speciation analysis confirms these deposits to be ferric ($Fe^{3+}$) iron. The function of intranuclear iron in these cells remains unresolved, although both damaging and protective mechanisms are considered. This finding shows that STXM is a powerful label-free tool for the in situ, nanoscale chemical characterisation of both organic and inorganic intracellular components. Future applications are likely to shed new light on incompletely understood biochemical mechanisms, such as metal dysregulation and morphological changes to cell nucleoli, that are important in understanding the pathogenesis of Parkinson's.

Parkinson's disease is a major neurological condition globally, yet still lacks a cure or effective treatment whilst the underlying causes remain unresolved[1]. Many characteristic Parkinson's symptoms (including tremor, rigidity and bradykinesia) are caused by brain dopamine deficiency, arising from the drastic loss of dopamine-producing neurons in the substantia nigra. These vulnerable dopaminergic neurons are known to accumulate excess iron in the disease state[2]. Whilst iron is essential for normal brain function, perturbations to normal iron homoeostasis may increase the redox burden on cells through excessive participation in electron transfer reactions and the associated generation of toxic free radicals, potentially contributing to cell death[3-5]. Neuromelanin-iron complexes are considered to be the major iron phase in melanised nigral neurons in normal individuals[6]. However, additional intracellular iron species may also be present, particularly in Parkinson's nigra where neuronal iron is elevated.

As well as iron accumulation, reduced nucleolar size and damage to cell nucleoli have been observed in dopaminergic neurons in Parkinson's[7]. The nucleolus, the largest sub-compartment of the cell nucleus, is the site of ribosomal ribonucleic acid (rRNA) transcription, RNA processing and ribosome subunit assembly in eukaryotic cells[8]. The nucleolus is also an essential intracellular stress sensor and plays a key role in cellular response to stress[9,10]. This raises intriguing questions about how this organelle responds to neuronal iron accumulation. However, little attention has previously been paid to the role of the nucleolus in relation to iron transport and accumulation[11].

Iron accumulation in cell nuclei has previously been found in glial cells of the hippocampus in Alzheimer's disease[12], and in neurons from the cerebral cortex in cases of neuroferritinopathy[13]. The accumulation of iron within neuronal nuclei has been implicated as a pathological event[13-15]. Perl's iron staining has been applied to demonstrate the presence of ferric iron within nucleoli in both rat and human neurons, suggesting that nucleolar iron may indeed contribute to peroxidation processes and damage to the cell nucleus[16]. One study has demonstrated the co-localisation of iron and

[1]School of Engineering, Library Road, University of Warwick, Coventry, CV4 7AL, UK. [2]School of Pharmacy and Bioengineering, Guy Hilton Research Centre, Thornburrow Drive, Keele University, Staffordshire, ST4 7QB, UK. [3]School of Life Sciences, Gibbet Hill Campus, University of Warwick, Coventry, CV4 7AL, UK. [4]Newcastle Brain Tissue Resource, Institute of Neuroscience, Newcastle University, Newcastle-upon-Tyne, NE4 5PL, UK. [5]Department of Chemistry, Library Road, University of Warwick, Coventry, CV4 7AL, UK. ✉e-mail: Jake.Brooks@warwick.ac.uk

**Fig. 1 | Correlative mapping of cell nucleoli in adjacent 500 nm sections of Parkinson's SNc.**
**a** Melanised neuron stained for RNA, **b** protein map of same cell shown in (**a**) in adjacent unstained tissue section, blue and orange arrows mark the nuclear membrane and nucleolus, respectively, **c** off-peak image at 530 eV shows neuromelanin distribution and the cell nucleolus due to their relatively high optical density, **d** iron map, where the red arrow marks the position of intranuclear iron deposit, yellow asterisks mark the positions of iron containing blood vessels, **e** composite map showing protein (blue), density-related contrast (white) and iron (red), **f** iron $L_{2,3}$-edge spectrum from intranuclear iron deposit shown in (**e**), demonstrating the presence of ferric iron (see reference spectra in Fig. 2).

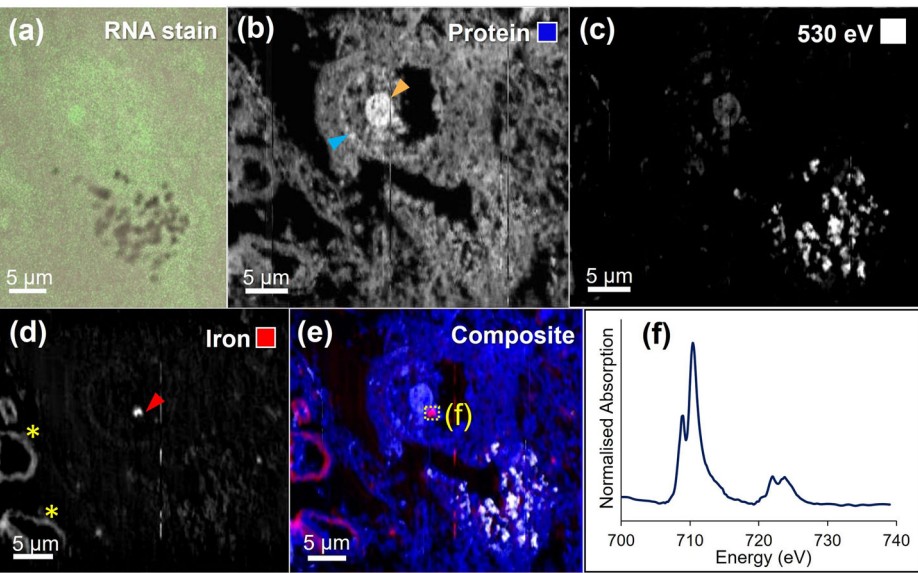

aluminium in neuronal nuclei of the human Alzheimer's brain, with the highest levels of both aluminium and iron measured inside the nucleolus[17]. A more recent study conducted by Lemelle et al. used synchrotron X-ray fluorescence (XRF) mapping to detect iron loading in the nucleoli of nigral neurons in a rat model of Parkinson's. Notably they commented that "*the possibility that the presence of Fe in the nucleoli might be an in vivo property of dopaminergic neurons calls for further investigation*"[18].

Regarding the detection and analysis of intranuclear iron, synchrotron x-ray spectromicroscopy has exceptional scope for label-free investigation of trace metal distributions at an intracellular level, owing to the capacity for combining excellent sensitivity and specificity with nanoscale spatial resolution. Whilst the use of stains or chemical fixatives is known to alter significantly native tissue chemistry[19], synchrotron techniques can be applied without the requirement for staining or sample ablation. By employing synchrotron x-ray spectromicroscopy techniques based on the soft x-ray regime, metal distributions can be visualised with the simultaneous characterisation of the organic tissue structures that determine the anatomy of a region. Scanning Transmission X-ray Microscopy (STXM) uses a soft x-ray approach and has the capacity to combine imaging and spectral analysis of biological samples[20–24]. STXM can be used to generate image contrast based on distinct spectral features. This means that mapping chemical element distributions can be further refined to map the respective spatial distributions of specific chemical species. For example, STXM can be used to probe for specific types of bonds or ions, enabling distinction between different proteins or metal ion oxidation states. We have previously applied STXM to mapping of species in organic intracellular components[20,22], and to investigate interactions between metal ions and amyloid-beta aggregates in vitro[25–27] and ex vivo[24,28]. This approach has described the biochemical environment and identified possible metal:protein interactions in key areas of disease pathology.

The aim of this study was to apply a unique, label-free STXM approach to investigate the iron distribution and chemical speciation within the nuclei of vulnerable melanised neurons in Parkinson's substantia nigra. This approach has allowed us to identify the presence of multiple iron foci, and to characterise their chemical and mineral properties, in melanised nigral neurons.

## Results

Figure 1 shows correlative RNA staining (elevated in cell nucleoli) and oxygen K-edge STXM analysis of a melanised neuron in a 500 nm thick section of Parkinson's substantia nigra pars compacta (SNc). STXM mapping at the oxygen K-edge, using the C = O peak at 532 eV, revealed the tissue ultrastructure. Mapping whole cells at 200 nm spatial resolution

permitted intracellular structures to be clearly visualised. Melanised neurons were identified in the tissue sections by acquiring off-resonance images at 530 eV, away from the 532 eV protein peak, where neuromelanin granules displayed elevated contrast due to neuromelanin's higher optical density at this energy relative to the surrounding neuropil (see Fig. 1c) as described previously[21]. Cell nuclei were identified by imaging at 532 eV due to increased contrast at the membrane of the nucleus, as shown in Fig. 1b. The cell nucleolus, enveloped within the nuclear membrane, was characterised by dramatically elevated contrast at the protein peak (Fig. 1b). Correlative RNA staining in an adjacent tissue section confirmed the identity of the cell nucleolus (Fig. 1a).

A concentrated intranuclear iron deposit (ca. 1 μm in diameter) was identified at the perimeter of the nucleolus (Fig. 1d). To determine the oxidation state of the intranuclear iron deposit, x-ray spectromicroscopy was performed over the iron $L_{2,3}$-edge (700–740 eV). Iron deposits of differing oxidation state show characteristically different spectral features, as illustrated in the reference spectra in Fig. 2. At the $L_{2,3}$-edge, ferric ($Fe^{3+}$) materials display principal x-ray absorption features at 709.5 and 723 eV, with smaller peaks at 708 and 721 eV. In contrast, ferrous ($Fe^{2+}$) and zero–oxidation state metallic ($Fe^{0}$) phases display principal features at 708 and 721 eV[29]. Spectroscopy analysis confirmed that the deposit was consistent with ferric ($Fe^{3+}$) iron.

STXM mapping of additional melanised neurons from the same Parkinson's case also revealed concentrated iron deposits, 1–1.5 μm in diameter, again located at the perimeter of the cell nucleolus (see Figs. 3 and 4). Spectroscopy analysis demonstrated that all measured iron deposits associated with the cell nucleolus were ferric ($Fe^{3+}$) iron, with very similar spectra acquired for each deposit, suggesting that iron was consistently present in the same form. In contrast, variation in oxidation state was observed for iron associated with neuromelanin granules (Fig. 3f), with some areas exhibiting enhanced ferrous ($Fe^{2+}$) absorption features, as we previously reported for neuromelanin-associated iron in Parkinson's[21]. Repeat stack measurements over the iron $L_{3}$-edge (with equivalent dwell time and aperture size) were performed over an iron-loaded blood vessel in close proximity to a cell of interest (Fig. 5). X-ray absorption spectra confirmed the presence of ferric iron associated with the blood vessel, and with repeat scans there was no evidence of beam-induced changes to the iron oxidation state.

## Discussion

The present study demonstrates a novel application of this label-free STXM characterisation method to characterise iron deposits inside the nuclei of vulnerable melanised neurons in Parkinson's substantia nigra. This work demonstrates that STXM is a powerful tool for analysis of both organic and

inorganic intracellular features. Neuromelanin clusters, nuclei and nucleoli were all visualised within melanised neurons in human tissue, without requirement for chemically-disruptive aldehyde fixatives or staining. With metal dysregulation widely reported as a feature of numerous neurode-

generative conditions[5,30,31], this work demonstrates the value of using STXM to interrogate the unresolved biochemistry underpinning these disorders.

In the single Parkinson's case studied here, iron deposits were shown to associate with the edge of the cell nucleolus in nigral neurons. Since iron can mediate DNA damage, it has been suggested that there may be an evolutionary advantage to keeping iron outside the nucleus in the healthy brain[11]. This hypothesis is supported by observations of intranuclear iron in disorders characterised by iron dysregulation, including Alzheimer's[12] and neuroferritinopathy[13]. Yumoto and co-workers reported markedly increased iron levels in the nuclei of nerve cells measured in Alzheimer's brain relative to age-matched controls, with the authors hypothesising that iron binding to nuclear(n)DNA may contribute to neurodegeneration by inducing oxidative nDNA damage and simultaneously inhibiting repair of damaged nDNA[17]. Further, Korzhevskii and co-workers reported the accumulation of non-haem iron in the nucleoli of melanised nigral neurons from two cases initially classified as neurologically normal. The two cases were later excluded from this classification due to prolonged illness of the subjects, leading the authors to suggest iron accumulation in the nucleoli as a pathological event[15].

However, the potential for iron to play a constructive, physiological role inside the nucleus cannot be excluded. In vitro models of dopaminergic cells exposed to an excess of iron have shown evidence of iron accumulation in the cytosol and neurite outgrowths, but crucially not inside the cell nuclei[32,33]. This suggests that the intranuclear iron observed in the present study may be a physiological response to the specific needs of the cell.

Interaction between iron regulatory proteins (IRP1 and IRP2) and iron responsive elements (IREs) present in the untranslated region of certain mRNAs is central to intracellular iron homoeostasis[34,35]. IRPs have RNA-binding properties that depend on the presence of a 4Fe-4S cluster[34]. Whilst IRPs are typically regarded as cytosolic proteins, nuclear localisation of IRP1 has been observed in iron-replete cells, suggesting a cell-specific response mediated by an iron-dependent mechanism[34].

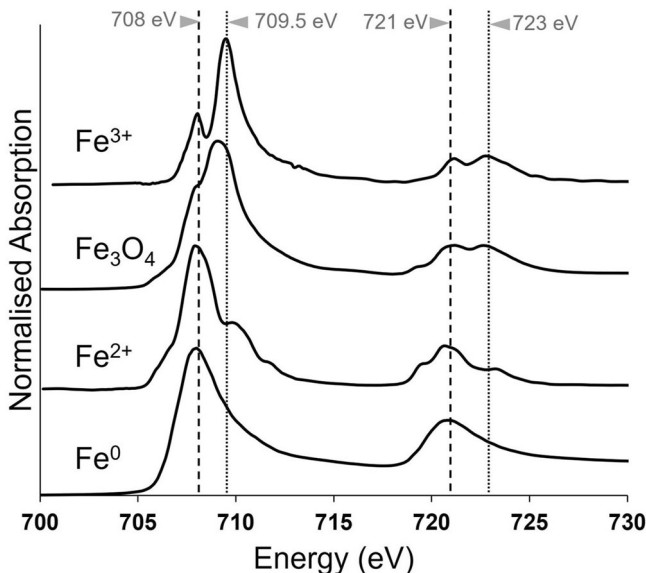

**Fig. 2 | Iron L$_{2,3}$-edge reference x-ray absorption spectra from iron of varying oxidation states.** Fe$_3$O$_4$ contains both Fe$^{2+}$ and Fe$^{3+}$. The dashed line at 708 eV and dotted line at 709.5 eV mark the principal absorption energies for Fe$^{2+}$ and Fe$^{3+}$ cations, respectively.

**Fig. 3 | STXM characterisation of melanised neuron in Parkinson's SNc. a** Protein map, showing cell membrane (yellow arrow), nucleus membrane (blue arrow) and nucleolus (orange arrow), **b** off-peak image at 530 eV showing distribution of neuromelanin, **c** iron map, **d** composite map showing protein (blue), neuromelanin (white), iron (red), **e** image of melanised cellular region highlighted in (**d**) at Fe$^{3+}$ peak, (**f**) iron L$_3$-edge spectra from neuromelanin clusters and extracellular region highlighted in (**e**). The dashed line at 708 eV and dotted line at 709.5 eV mark the principal absorption energies for Fe$^{2+}$ and Fe$^{3+}$ cations, respectively. **g** Iron L$_{2,3}$-edge spectrum from intranuclear iron deposit highlighted in (**c**).

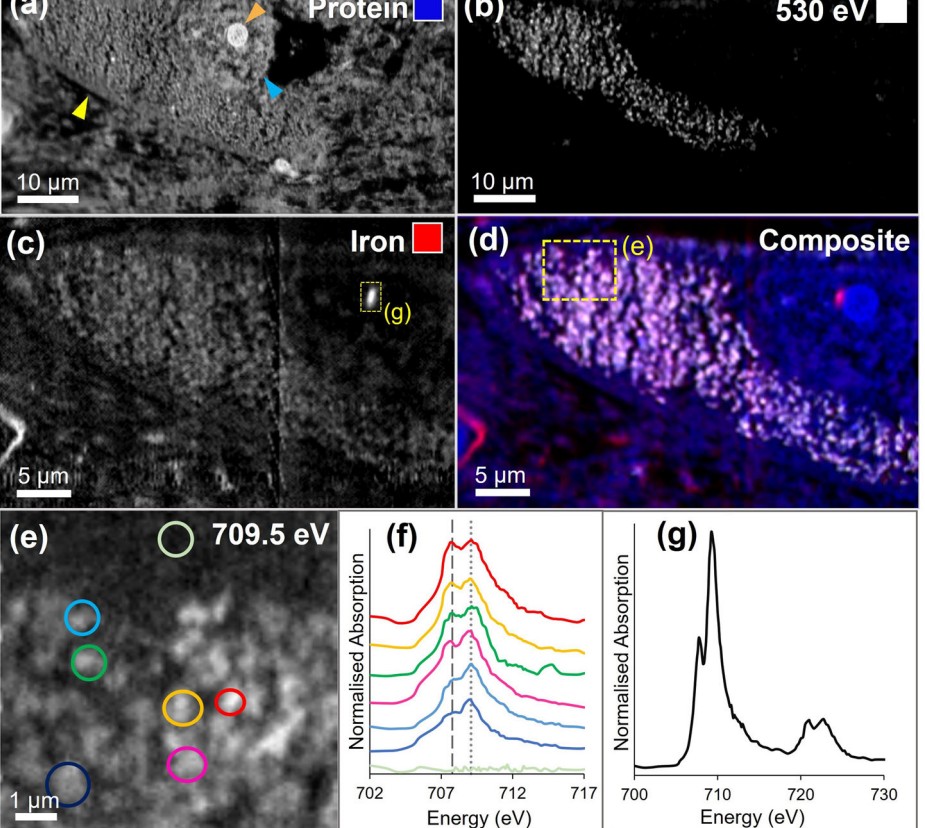

**Fig. 4 | STXM characterisation of melanised neuron in Parkinson's SNc. a** Protein map, (**b**) off-peak image at 530 eV showing distribution of neuromelanin, (**c**) iron map of inset nucleus region highlighted in (**a**), (**d**) iron $L_{2,3}$-edge spectrum from nuclear iron deposit associating with nucleolus highlighted in (**c**), (**e**) iron $L_{2,3}$-edge spectrum from second intranuclear iron deposit highlighted in (**c**).

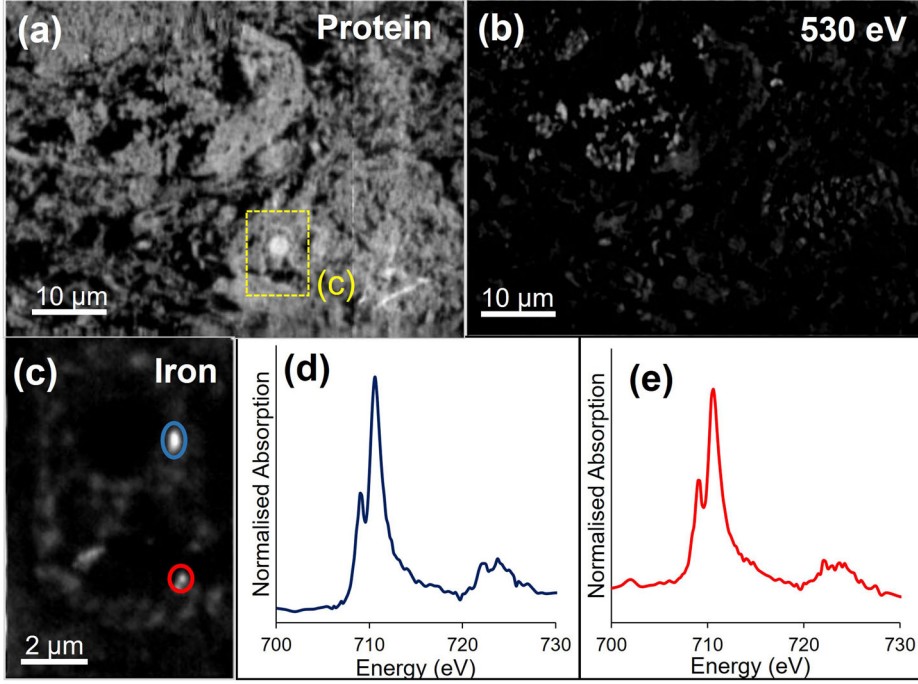

**Fig. 5 | Ensuring x-ray dose avoids beam-induced changes to x-ray absorption spectra.** (**a**) Iron map of blood-vessel in close proximity to neurons, (**b**) successive iron $L_3$-edge stack measurements repeated over the same iron-containing blood vessel highlighted in (**a**). Spectra were derived from the entire blood vessel.

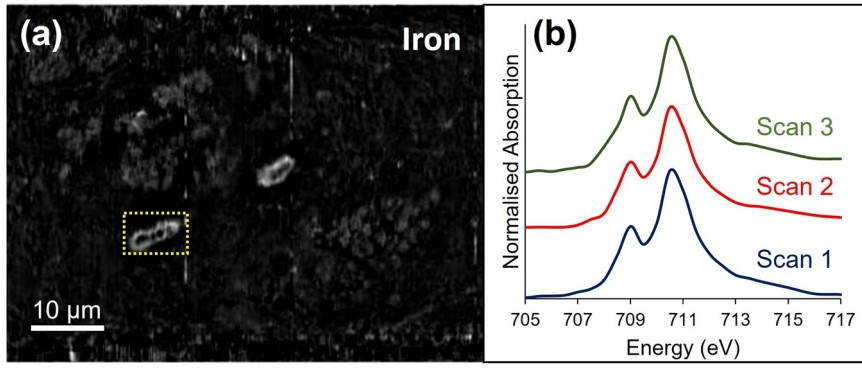

Early observations of protein-bound iron in the nucleoli of HeLa cells has led to the suggestion that the nucleolus may be an iron repository for iron-dependent DNA synthesis proteins[36]. In a more recent study, Quintana et al. observed that sub-nuclear iron-rich regions in Alzheimer's' disease glial cells can be attributed to ferritin deposits, speculating that ferritin may serve to protect DNA from oxidative damage by scavenging redox active iron and storing it as a redox-inactive form of $Fe^{3+}$[12]. Although ferritin has traditionally been considered as a cytoplasmic iron storage protein, numerous studies support a role for nuclear ferritin in human development and disease, specifically in the oxidative protection of DNA, and in transcriptional regulation[37–39]. Intranuclear inclusions known as Marinesco bodies and Roncorini rodlets have also been shown to contain non-haem iron in nigral neurons of neurologically-healthy individuals, suggesting the presence of iron-containing proteins in these inclusions[40]. Although it was not confirmed in the current study if iron was stored in ferritin, STXM was used to confirm the presence of ferric iron associated with cell nucleoli, which would be consistent with a hypothesis attributing this iron to ferritin. A further possible explanation for iron associating with cell nucleoli is that iron plays a role in rRNA biosynthesis (as yet ill-defined), postulated by Roschzttartz et al. upon discovery of iron accumulation within the nucleoli of pea plant embryonic cells[41].

The results obtained here provide proof-of-concept for the existence of intranuclear iron associating with the cell nucleolus in melanised nigral neurons, using tissues from a confirmed Parkinson's case. The function of intranuclear iron in these cells remains unresolved, although both damaging and protective mechanisms are possible. Having demonstrated this approach, this discovery now warrants extended investigation, including studies of Parkinson's cases across a spectrum of ages, and neurologically healthy controls, to determine to what extent the abundance of intranuclear iron deposits observed here might be a pathological feature of Parkinson's. Moreover, links between abnormal nucleolus morphology in Parkinson's and metal dysregulation as a potential driver of oxidative stress have yet to be fully investigated. This work demonstrates that STXM is ideally suited to chemical characterisation of organelles such as the nucleolus, and indicates the value in the future use of STXM to explore the potential for correlation between nucleolus size/morphology and alterations in the redox properties of intracellular metal ions.

## Materials and methods
### Tissue preparation
Frozen, chemically-unfixed SNc tissue from a confirmed Parkinson's case was obtained from the Newcastle Brain Tissue Resource and prepared and analysed under ethical approvals 07/MRE08/12 and REGO-2018-2223.

Substantia nigra tissue was cut into cubes approximately $8mm^3$ in volume using a non-metallic knife to prevent metal contamination, and dehydrated using an ethanol series (40–100% dry). Following dehydration, tissue cubes were embedded in an aliphatic epoxy resin composed of an equimolar mixture of trimethylolpropane triglycidyl ether and 4,4′-methylenebis-(2-methylcyclohexylamine), purchased from Sigma Aldrich (Dorset, UK). Tissue samples were initially immersed in 75% dry ethanol, 25% resin. The resin proportion was increased in increments of 25% every two hours with mixing, with two further hourly changes of 100% resin, and then curing overnight in 1.5 ml moulds at 60 °C.

For soft x-ray spectromicroscopy (STXM) experiments, 500 nm-thick sections of resin-embedded substantia nigra tissue were cut using a Reichert-Jung ultra-cut microtome, operating with a diamond blade (DiATOME Ultra 45°). Consecutive sections were obtained to enable correlative immunostaining and STXM analysis of the same cells. Sections intended for RNA staining were dried onto glass microscope slides. Sections to be analysed by STXM were air-dried onto copper TEM grids, with no dyes or contrast agents applied.

### Scanning transmission x-ray microscopy

STXM experiments were performed at Diamond Light Source beamline I08 (Oxfordshire, UK). Microscopy images were obtained by tuning the incident x-ray beam to a target energy, raster scanning the sample using a focussed beam (size ca. 50 nm) and recording the transmitted x-ray intensity.

Speciation maps showing the nanoscale distribution of particular elements or chemical states were generated by collecting paired images: a peak image at the energy corresponding to a feature of interest (e.g. the principal $Fe^{3+}$ $L_3$-edge peak at 709.5 eV), and an off-peak image a few eV below this feature. Raw images were first converted to optical density using background areas of the resin devoid of tissue and aligned using image cross-correlation analysis to account for beam drift with changing energy. The off-peak image was then subtracted from the peak image providing a speciation map devoid of artefacts (for example score marks left by the sectioning blade). Speciation maps were created for features at the oxygen K-edge (520−545 eV) and the iron $L_3$-edge (700−716 eV). Spatial resolution acquisition parameters for speciation maps ranged from 200 nm for large maps of whole cells, to 80 nm for small inset maps of intracellular regions.

Iron $L_{2,3}$-edge x-ray absorption spectra were obtained by collecting a series of images (stack) at multiple x-ray energies across the absorption edge. Collected absorption spectra provided detailed information regarding the oxidation state of the measured iron deposits. Image stacks were performed over localised intracellular regions of interest with spatial resolution between 75 and 100 nm. Image stacks were converted to optical density (thereby normalising the resultant x-ray absorption spectra) and aligned, as described for the speciation maps. This form of x-ray spectromicroscopy enabled spectra to be generated from each pixel of an image, allowing the iron oxidation state from highly localised regions of interest to be determined.

STXM data processing was performed using the aXis 2000 software package (http://unicorn.mcmaster.ca/aXis2000.html). To remove interference patterns caused by the injection of electrons into the synchrotron storage ring (top-up), a bandpass image filter was applied using ImageJ. Pseudo-coloured composite images were created by converting greyscale images to false colour, before recombining the images as an overlay using ImageJ.

### Correlative RNA staining and laser scanning fluorescence microscopy

Staining for RNA was used to locate cell nucleoli in tissue sections. All solutions were pipetted onto the sections as 5 µL droplets. Tissue sections were washed three times with phosphate-buffered saline (PBS) to remove any contaminants adhering to the surface of the sections. Epitope retrieval was performed using a buffer of 10 mM sodium citrate and 0.05% Tween 20, adjusted to pH 6.0 with HCl and heated to 90 degrees centigrade. The buffer was applied to each section for 20 min, before excess was removed using filter paper. A blocking solution (1% BSA, 0.4% Triton 100X in PBS) was applied for 20 minutes. RNA staining was performed using SYTO RNA-Select green fluorescent cell stain (ThermoFisher), diluted 1:10000 and applied to tissue sections for approximately 1 hour. Stained tissue sections were washed with PBS before coverslip mounting using Vectashield (Vector Laboratories).

Fluorescence imaging was performed using a Zeiss LSM 880 confocal microscope, equipped with three photomultiplier detectors and an ultra-sensitive Gallium Arsenide Phosphide (GASP) detector. Fluorescence was stimulated by scanning with a laser of wavelength 490 nm. Images were acquired in fluorescence and bright field modes using Zen Black software.

### Statistics and reproducibility

The work presented constitutes an exploratory, proof of concept, study to evaluate the utility of synchrotron x-ray spectromicroscopy for intranuclear iron characterisation. As such, a single case ($n = 1$) was used to demonstrate this. Intranuclear iron was consistently shown to be in the same ferric oxidation state across multiple neurons measured. Repeat scans were also used to verify scanning parameters, with no evidence of beam-induced changes to the iron oxidation state. This confirmed measurements to be reproducible.

### Reporting summary

Further information on research design is available in the Nature Portfolio Reporting Summary linked to this article.

### Data availability

The data that support the findings in this study will be available in the Warwick Research Archive Portal (WRAP) repository at https://wrap.warwick.ac.uk/185993.

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

## Acknowledgements

This work was supported by EPSRC grants EP/N033191/1, EP/N033140/1. JB has received funding support from an EPSRC Doctoral Training Award (EP/N509796/1), Warwick-Wellcome QBP Covid Relief Funding, an Alzheimer's Research UK small ECR Grant, a Warwick-Wellcome Translational Fellowship (WT-219429/Z/19/Z), and a Race Against Dementia Fellowship (ARUK-RADF2022B-008) funded by the Joseph and Lillian Sully Foundation in collaboration with Alzheimer's Research UK. J.E. has received funding support from an Alzheimer's Research UK Early Career Researcher Bridge Fund (ARUK-ECRBF2022A-017), an Alzheimer's Research UK Midlands Network Small ECR Grant, and the Keele University Faculty Research Fund. E.H. holds a Race Against Dementia Fellowship funded by the Barbara Naylor Foundation, in collaboration with Alzheimer's Research UK. Human brain tissue used in this study was analysed in accordance with the Declaration of Helsinki under the remit of ethical approvals REGO-2018-2223 from the BSREC at University of Warwick, and approval 07/MRE08/12 from the North West Haydock Ethics Committee. Human tissue was obtained with informed consent from Newcastle Brain Tissue Resource, with thanks to Debbie Lett for assisting with tissue provision. All ethical regulations relevant to human research participants were followed. We thank Diamond Light Source for access to beamline I08 (proposals MG29042 and MG24534) and Dr T. Araki, Dr M. Kazemian Abyaneh and Dr B. Kaulich for technical assistance at the beamline. We also thank the Advanced Light Source for access to beamline 11.0.2 to collect synchrotron data presented in this study.

## Author contributions

Conceptualisation: Jake Brooks; Methodology: Jake Brooks, James Everett, Neil Telling, Joanna Collingwood. Formal analysis and investigation: Jake Brooks, James Everett, Emily Hill, Kharmen Billimoria, Neil Telling; Writing - original draft preparation: Jake Brooks; Writing - review and editing: Jake Brooks, James Everett, Emily Hill, Kharmen Bilimoria, Christopher Morris, Peter Sadler, Neil Telling, Joanna Collingwood; Funding acquisition: Jake Brooks, James Everett, Peter Sadler, Neil Telling, Joanna Collingwood; Resources: Emily Hill, Christopher Morris, Joanna Collingwood; Supervision: Joanna Collingwood

## Competing interests

The authors declare no competing interests.
