## [Peer review file · Communications Biology]

Reviewers' comments:

Reviewer #1 (Remarks to the Author):

In this paper titled "Nanoscale synchrotron x-ray analysis of intranuclear iron in melanised neurons of the Parkinson's substantia nigra", Jake Brooks et al. focuses on the accumulation of iron within the neurons of the substantia nigra in the human brain, which is a characteristic observed in Parkinson's disease (PD). The researchers used scanning transmission x-ray microscopy (STXM) to investigate the presence and nature of iron deposits in these neurons, providing sound evidence to identify the presence of multiple iron foci and their chemical properties for the first time. From the reviewer's perspective, this manuscript is suitable for publishing in *Communications Biology* with some minor revisions by answering the following questions.

1. In tissue preparation section, samples were all embedded in resin prior to ultramicrotome. Do you expect the iron oxidation state to remain intact during the embedding process? Have you considered the option to prepare and characterize the samples in frozen state?

2. In STXM setup, could you explain how you can use a 50um probe to achieve 200nm spatial resolution maps? Additionally, could you also provide the dwell time, estimated dosage and energy resolution?

3. In Fig 3 e&f, could you provide another baseline spectra on adjacent dark region of the iron map?

4. Do you have an explanation why neuromelanin region shows iron traces in Fig 3 but not in Fig. 2?

Reviewer #2 (Remarks to the Author):

The manuscript by Brooks et. al., describes a novel microscopy approach (STXM) to study sub-cellular Fe distribution in neuromelanin containing neurons in a case of Parkinson's disease. The novelty of the methods used should not be understated, and the data is of high value, revealing new information that is difficult to obtain by other methods.

Weaknesses of the paper are that a greater number of replicates were not analysed, and control neurons from healthy individuals were also not analysed. The reviewer acknowledges that obtaining access to such samples, and sufficient access to the analytical equipment is prohibitive though.

Although only a limited number of cells were imaged, the association of Fe with the nucleolus and RNA is very interesting. As mentioned above, the analytical capability to directly resolve sub-cellular Fe distribution, and to visualise its accumulation next to the nucleolus is very novel. I feel the identification of the Fe enrichment next to the nucleolus is a significant finding, and will enable other studies to further investigate this topic. I am however, surprised that the manuscript does not contain a discussion on iron responsive elements (IREs) / iron responsive proteins (IRPs) and how this may relate to nucleolar Fe content – I think the discussion would

benefit if a paragraph was added on this topic.

The following statement on page 4 of the PDF should have a reference. "Whilst use of stains or chemical fixatives is known to alter significantly native tissue chemistry, synchrotron techniques can be applied without requirement for staining or sample ablation"

Two suitable references that the authors could choose to support the above statement are listed below, however no obligation to use those specific references, there are multiple publications describing the elemental alterations that occur to tissue as a consequence of chemical fixation.

Chwiej, Joanna, Magdalena Szczerbowska-Boruchowska, Marek Lankosz, Sławomir Wojcik, Gerald Falkenberg, Zdzisław Stegowski, and Zuzanna Setkiewicz. "Preparation of tissue samples for X-ray fluorescence microscopy." *Spectrochimica Acta Part B: Atomic Spectroscopy* 60, no. 12 (2005): 1531-1537.

Hackett, Mark J., James A. McQuillan, Fatima El-Assaad, Jade B. Aitken, Aviva Levina, David D. Cohen, Rainer Siegle et al. "Chemical alterations to murine brain tissue induced by formalin fixation: implications for biospectroscopic imaging and mapping studies of disease pathogenesis." *Analyst* 136, no. 14 (2011): 2941-2952.

Presumably a large portion of the Fe in the blood vessel in Figure 5 is heme-bound Fe. Are there any spectral features in Figure 5 that help confirm the speciation of Fe as heme (e.g., hemoglobin).

Lastly, the overall "tone" of the manuscript is that Fe accumulation is "bad", and that the Fe enrichment near the nucleolus is a component of PD pathology. However, as the authors haven't studied healthy neurons this remains to be shown. Further, although Fe accumulation can be "bad", Fe is essential for cell function, especially neuronal function. I feel like the discussion could be better balanced to acknowledge that the Fe accumulation near the nucleolus might not be a consequence of pathology and could indeed be involved in healthy function of the neuron.

We thank both reviewers for their detailed consideration of this manuscript and for their highly supportive feedback. A point-by-point response to all reviewer comments is provided below.

Reviewer 1

1. *"In this paper titled "Nanoscale synchrotron x-ray analysis of intranuclear iron in melanised neurons of the Parkinson's substantia nigra", Jake Brooks et al. focuses on the accumulation of iron within the neurons of the substantia nigra in the human brain, which is a characteristic observed in Parkinson's disease (PD). The researchers used scanning transmission x-ray microscopy (STXM) to investigate the presence and nature of iron deposits in these neurons, providing sound evidence to identify the presence of multiple iron foci and their chemical properties for the first time. From the reviewer's perspective, this manuscript is suitable for publishing in Communications Biology with some minor revisions by answering the following questions."*

We thank the reviewer for their positive comments concerning the evidence presented in this manuscript.

2. *"In tissue preparation section, samples were all embedded in resin prior to ultramicrotome. Do you expect the iron oxidation state to remain intact during the embedding process? Have you considered the option to prepare and characterize the samples in frozen state?"*

We acknowledge that preserving the integrity of the iron chemistry is fundamental to this study. In prior work, we have performed extensive method development to ensure that the observations reported are robust and a feature of the system of interest, as opposed to an artefact of the resin embedding. The detail of this work was incorporated into our first publication concerning STXM analysis of iron in mammalian tissues (Telling et al., Cell Chemical Biology, 2017). We demonstrated that iron standards prepared using the same embedding series show no alteration in oxidation state when examined using STXM. On this basis, we expect the oxidation state to remain intact.

Analysis of frozen-hydrated tissue sections represents a significant technical advancement for STXM, and is not currently available at Diamond I08, where data for this study were collected. Logistical challenges associated with human brain tissue samples constitute an additional barrier. However, exciting ongoing development of cryogenic measurement capabilities at many synchrotron beamlines (see e.g. Leontowich et al. Review of Scientific Instruments, 2018) will soon create opportunities for the examination of vitrified biological specimens. In addition to the preservation of native sample (bio)chemistry and ultrastructure during sample preparation, performing STXM under cryogenic conditions also protects against X-ray damage induced by sample measurement at high X-ray doses,

allowing for increased tolerances to X-ray radiation (Hitchcock, et al., *Microscopy and Microanalysis*, 2020), which will improve data quality. We are currently working towards cryogenic-analysis in partnership with synchrotron beamline staff and hope to be in a position to trial STXM measurements under cryogenic conditions in the near future. We discuss the implementation of cryogenic STXM measurements and the benefits thereof, in our recent papers: Everett et al., *JESRP*, 2023 and Everett et al. *ACS Chemical Neuroscience*, 2024).

3. *"In STXM setup, could you explain how you can use a 50um probe to achieve 200nm spatial resolution maps? Additionally, could you also provide the dwell time, estimated dosage and energy resolution?"*

We thank the reviewer for highlighting this error in the text. The diameter of the focussed beam is approximately 50 nm, not 50 μm . This has been corrected in the revised manuscript (line 263).

The dwell time for all STXM maps was fixed at 10 ms.

For each presented x-ray absorption spectrum, energy resolution is typically not fixed across the entire spectrum. Energy resolution is increased (to a maximum achievable resolution of 0.1 eV) over the features of interest to resolve accurately the shape and position of the peaks, and then reduced (to a minimum of 1 eV) in the pre- and post-edge ranges. This makes it possible to acquire sufficient detail to interpret metal ion speciation from x-ray absorption spectra, whilst also permitting efficient use of limited synchrotron time. Full stack parameters are provided for each spectrum below. These stack parameters will also be accessible via the accompanying dataset upon publication.

Figure 1(f): 700-704 eV in 1 eV steps; 704.5-706 eV in 0.5 eV steps; 706.2-713 eV in 0.2 eV steps, 713.25-715.5 eV in 0.25 eV steps; 716-719 eV in 0.5 eV steps, 719.25-726 eV in 0.25 eV steps, 726.5-729 eV in 0.5 eV steps, 730-740 in 1 eV steps.

Figure 2:

Fe(0) spectrum: Fixed at 0.1 eV steps for entire spectrum.

Fe²⁺ spectrum: 700.5-703.15 eV in 0.5 eV steps; 703.35-713.1 eV in 0.15 eV steps; 713.45-718.25 eV in 0.4 eV steps; 718.35-735.15 eV in 0.15 eV steps.

Fe³⁺ spectrum: 700.65-703.65 eV in 1 eV steps; 704.75-713.55 eV in 0.1 eV steps; 714.15-719.15 eV in 0.5 eV steps; 719.95-733.65 eV in 0.3 eV steps.

Fe₃O₄ spectrum: Fixed at 0.1 eV for entire spectrum

Figure 3 (f, g): 700-705 eV in 1 eV steps; 705.5-707 eV in 0.5 eV steps; 707.125-713 eV in 0.125 eV steps; 713.25-715.5 eV in 0.25 eV steps; 716-719 eV in 0.5 eV steps; 719.25-726 eV in 0.25 eV steps; 726.5-729 eV in 0.5 eV steps; 730-740 eV in 1 eV steps.

Figure 4 (d, c): 700-704 eV in 1 eV steps; 704.5-706 eV in 0.5 eV steps, 706.2-713 eV in 0.2 eV steps, 713.25-715.5 eV in 0.25 eV steps, 716-719 eV in 0.5 eV steps, 719.25-726 eV in 0.25 eV steps, 726.5-729 eV in 0.5 eV steps, 730-740 in 1 eV steps.

Figure 5 (spectra 1 – 3): 700-705 eV in 1 eV steps; 705.5-714 eV in 0.5 eV steps; 715-718 in 1 eV steps.

We acknowledge the reviewer's request for estimated dosage, and understand that x-ray doses during sample measurement must be carefully controlled to ensure these doses do not significantly alter the chemistry of the material being examined.

However, in practice, it is extremely difficult to achieve any meaningful estimate of x-ray dosage due to intrinsic variability in the estimation of parameters that contribute to dose calculation, and the substantial variability in x-ray dose damage thresholds for differing elements and differing specimen preparations. X-ray imaging dose (D) can be estimated from the total interaction cross-section as follows:

$$D = \frac{I_0 t E}{A} \left[\frac{\mu}{\rho} \right]$$

Where I_0 is the incident photon intensity, E is the photon energy, $\left[\frac{\mu}{\rho} \right]$ is the x-ray mass attenuation coefficient, A is the sample area, and t is the exposure time. As described in Jones et al. Analytical Chemistry, 2017, whilst this dose estimate considers all of the modalities of energy transfer into the specimen, it makes no reference to the particulars of the interaction between the x-ray probe and the specimen: there is no distinction between dose that is deposited as heat, that which results in core or valence ionisation, or that which results in targeted bond-breaking within a system. This estimated dose calculation, carries at best an uncertainty order of 50%, where a specimen composition is unequivocally known. Even in homogenous specimens, small changes in the assumption of empirical formula can result in significant effects on dose estimates.

Here, we examined a complex sample of biological origin where the complete composition and structure cannot be fully determined; it is therefore impossible to provide an accurate dose estimate. Furthermore, our STXM measurements showed sample composition to be entirely heterogeneous, displaying variation at a nanoscale (i.e. pixel by pixel) level. Therefore, an example of a $5 \mu\text{m}^2$ scanning area, measured at 50 nm spatial resolution, would potentially provide 10,000 different estimates of x-ray dose. Even making the most

basic assumption that these samples were composed of two constituents: resin and tissue (of empirically-determined composition), the relative contribution of each of these constituents to the x-ray mass attenuation coefficient could vary from pixel to pixel.

By extension, if we were to provide a coarse x-ray dose estimate per scan (assuming uniform dose for each pixel of the scanning area), the heterogeneous nature of our sample material would mean the effect of this x-ray dose would also vary pixel by pixel.

Therefore, suitable x-ray dose(s) are empirically-tailored for each individual experiment dependent on the needs of the specific study and scientific questions under investigation.

We have an established track record in diligently assessing x-ray dose effects and controlling scanning parameters to match the needs of our experiments. Examples of this seen in the following:

Everett et al. *Science Advances*. 7,eabf6707(2021) Supplementary Materials text and Figures S11-S13; Brooks et al. *Angewandte Chemie*. 59, 11984-11991 (2020) Figure S7; Everett et al. *Scientific Reports*. 10, 10332 (2020) Figure S3.

We further demonstrate the successful optimisation of our scanning parameters (dosage) in Figure 5 of the current manuscript, where we demonstrate that the oxidation state of iron deposits within the sample material is maintained over three successive scans of the same area; demonstrating that the applied x-ray dosages were not sufficient to induce photoreduction of the iron.

4. *"In Fig 3 e&f, could you provide another baseline spectra on adjacent dark region of the iron map"*

This has now been added to Figure 3 e&f as requested (line 147 and shown below). The spectrum from the adjacent, extracellular region shows no detectable iron signal.

Figure 1: STXM characterisation of melanised neuron in Parkinson's SNc. **(a)** Protein map, showing cell membrane (yellow arrow), nucleus membrane (blue arrow) and nucleolus (orange arrow), **(b)** off-peak image at 530 eV showing distribution of neuromelanin, **(c)** iron map, **(d)** composite map showing protein (blue), neuromelanin (white), iron (red), **(e)** image of melanised cellular region highlighted in (d) at Fe^{3+} peak, **(f)** iron L_3 -edge spectra from neuromelanin clusters and extracellular region highlighted in (e). The dashed line at 708 eV and dotted line at 709.5 eV mark the principal absorption energies for Fe^{2+} and Fe^{3+} cations, respectively. **(g)** Iron $L_{2,3}$ -edge spectrum from intranuclear iron deposit highlighted in (c).

5. "Do you have an explanation why neuromelanin region shows iron traces in Fig 3 but not in Fig. 2?"

In this example, the iron signal associated with the neuromelanin clusters is faint and was thresholded out of the original Figure 1. In the revised manuscript (line 111), the lower

threshold on the iron map in Figure 1d has now been adjusted to capture the diffuse iron signal – the composite image in Figure 1e has been updated accordingly. The updated figure is shown below.

Whilst neuromelanin has a widely recognised high affinity for iron, it should be noted that not all melanised neurons harbour an equivalent iron load. The range in neuronal iron load in substantia nigra has also been shown to be particularly pronounced in the Parkinson's state (Oakley et al., *Neurology*, 2007). Hence the variation in iron signal amongst neurons is not unexpected.

Figure 2: Correlative mapping of cell nucleoli in adjacent 500 nm sections of Parkinson's SNc. **(a)** Melanised neuron stained for RNA, **(b)** protein map of same cell shown in (a) in adjacent unstained tissue section, blue and orange arrows mark the nuclear membrane and nucleolus, respectively, **(c)** off-peak image at 530 eV shows neuromelanin distribution and the cell nucleolus due to their relatively high optical density, **(d)** iron map, where the red arrow marks the position of intranuclear iron deposit, yellow asterisks mark the positions of iron containing blood vessels, **(e)** composite map showing protein (blue), density-related contrast (white) and iron (red), **(f)** iron L_{2,3}-edge spectrum from intranuclear iron deposit shown in (e), demonstrating the presence of ferric iron (see reference spectra in Figure 2).

Reviewer 2

6. *"The manuscript by Brooks et. al., describes a novel microscopy approach (STXM) to study sub-cellular Fe distribution in neuromelanin containing neurons in a case of Parkinson's disease. The novelty of the methods used should not be understated, and the data is of high value, revealing new information that is difficult to obtain by other methods."*

We thank the reviewer for their positive feedback concerning the novelty of both the approach and value of the findings presented in this manuscript.

7. *"Weaknesses of the paper are that a greater number of replicates were not analysed, and control neurons from healthy individuals were also not analysed. The reviewer acknowledges that obtaining access to such samples, and sufficient access to the analytical equipment is prohibitive though"*

We appreciate that we had not included data from controls in this manuscript, and as such made no claims that findings of intranuclear iron made in this study are disease-specific. Nevertheless, we feel the topic is important in the context of better understanding the role of iron in both disease and in normal brain function and the findings presented along with the discussion material will stimulate further investigation (in full agreement with the reviewer's subsequent comment below).

We would also like to take this opportunity to present STXM data from two control neurologically-healthy control cases (see Figure 1 below), with intranuclear iron not observed in either case. However, we deliberately chose not to include these data in the manuscript as they imply a comparison between disease and control state that we cannot legitimately make with the available dataset. In addition to the small sample size, tissue sections were cut to sub-micron thickness (as required for STXM). Small iron deposits may therefore be present in the control samples but out of the plane of the tissue section. It is not anticipated that the advanced x-ray methods being applied here can be used for high-throughput studies (i.e. to enable group sizes >10), therefore direct comparison of intranuclear iron content in disease versus control state for a statistically significant number of neurons is currently unrealistic. This would require technical developments that might soon become feasible (as demonstrated by delivery of high-throughput synchrotron measurements for protein crystallography) but are beyond the scope of this particular study.

As supported by the reviewer comment below, the value in this approach is instead in determining the composition, speciation, and distribution of metal-rich deposits at the organelle level, whilst also capturing the anatomical context without requirement for staining, ablation, or for correlative analysis. This cannot be achieved with any other

technique and this manuscript serves to showcase that unique capability and applicability to unanswered questions in biology.

Figure 1: STXM characterisation of melanised neurons in neurologically-healthy control SNc: Case C1 **(a-b)** Case C2 **(c-d)**. **(a)** Protein map, nucleolus marked by an orange arrow **(b)** iron map of full area shown in (a), **(c)** single image at 705 eV, nucleolus marked by an orange arrow, **(d)** iron map of area shown in (c).

8. *“Although only a limited number of cells were imaged, the association of Fe with the nucleolus and RNA is very interesting. As mentioned above, the analytical capability to directly resolve sub-cellular Fe distribution, and to visualise its accumulation next to the nucleolus is very novel. I feel the identification of the Fe enrichment next to the nucleolus is a significant finding, and will enable other studies to further investigate this topic.”*

We thank the reviewer for recognising the value in showcasing the analytical capability and fully agree that it will enable future studies to further investigate this important topic.

9. *"I am however, surprised that the manuscript does not contain a discussion on iron responsive elements (IREs) / iron responsive proteins (IRPs) and how this may relate to nucleolar Fe content – I think the discussion would benefit if a paragraph was added on this topic."*

We thank the reviewer for their insightful suggestion to include this additional material and agree that it enriches the discussion.

As such, we have added the text below to the amended version of the manuscript (lines 201 – 206):

"Interaction between iron regulatory proteins (IRP1 and IRP2) and iron responsive elements (IREs) present in the untranslated region of certain mRNAs is central to intracellular iron homeostasis [1, 2]. IRPs have RNA-binding properties that depend on the presence of a 4Fe-4S cluster [1]. Whilst IRPs are typically regarded as cytosolic proteins, nuclear localisation of IRP1 has been observed in iron-replete cells, suggesting a cell-specific response mediated by an iron-dependent mechanism [1]."

[1] Gu, W., C. Fillebeen, and K. Pantopoulos, Human IRP1 translocates to the nucleus in a cell-specific and iron-dependent manner. *International Journal of Molecular Sciences*, 2022. 23(18): p. 10740.

[2] Pantopoulos, K., Iron metabolism and the IRE/IRP regulatory system: an update. *Annals of the New York Academy of Sciences*, 2004. 1012(1): p. 1-13.

10. *"The following statement on page 4 of the PDF should have a reference. "Whilst use of stains or chemical fixatives is known to alter significantly native tissue chemistry, synchrotron techniques can be applied without requirement for staining or sample ablation"*

We agree with the reviewer and have added the following reference to support this section of text (line 74).

*"Chwiej, J., Szczerbowska-Boruchowska, M., Lankosz, M., Wojcik, S., Falkenberg, G., Stegowski, Z. and Setkowicz, Z., 2005. Preparation of tissue samples for X-ray fluorescence microscopy. *Spectrochimica Acta Part B: Atomic Spectroscopy*, 60(12), pp.1531-1537."*

11. *"Presumably a large portion of the Fe in the blood vessel in Figure 5 is heme-bound Fe. Are there any spectral features in Figure 5 that help confirm the speciation of Fe as heme (e.g., hemoglobin)."*

Prior work by Hocking and colleagues (Hocking et al., Journal of the American Chemical Society, 2007) compares iron L-edge spectra for heme and non-heme iron compounds. Findings demonstrate that the iron L-edge spectrum from the example heme iron compound [Fe(tpp)(ImH)₂] decreases in total intensity and shifts by 0.1 eV to a lower energy compared to the non-heme reference.

This very subtle energy shift is lower than the minimum energy step used to collect spectra presented in Figure 5 of this manuscript. Therefore, whilst theoretically possible to distinguish between heme and non-heme sources using iron L-edge spectroscopy, this could not be concluded from the available dataset. We emphasise that the spectra presented in Figure 5 are used solely to validate the measurement parameters, and do not impact upon the conclusions drawn from the study.

12. *"Lastly, the overall "tone" of the manuscript is that Fe accumulation is "bad", and that the Fe enrichment near the nucleolus is a component of PD pathology. However, as the authors haven't studied healthy neurons this remains to be shown. Further, although Fe accumulation can be "bad", Fe is essential for cell function, especially neuronal function. I feel like the discussion could be better balanced to acknowledge that the Fe accumulation near the nucleolus might not be a consequence of pathology and could indeed be involved in healthy function of the neuron."*

We thank the reviewer for bringing this to our attention. In full understanding that iron enrichment could be constructive or destructive, we made considerable effort to balance the argument in the Discussion section of the manuscript.

Following the statement in the Discussion *"the potential for iron to play a constructive, physiological role inside the nucleus cannot be excluded"*, we have described several roles by which intranuclear iron could, as the reviewer correctly points out, be involved in healthy neuronal function. These include, as a repository for iron-dependent DNA synthesis proteins, to protect DNA from oxidative damage oxidation, and potential involvement in RNA biosynthesis.

With the addition of the paragraph above concerning possible intranuclear localisation of IRP (as suggested by the reviewer), we hope that the tone of the discussion section is adequately balanced.

REVIEWERS' COMMENTS:

Reviewer #3 (Remarks to the Author):

I feel that the authors have adequately addressed the points I previously raised. I have also read through the other reviewer comments, and I feel that the authors have adequately addressed those comments also.

Therefore I recommend publication, and congratulate the authors on a very nice study.

We thank both reviewers for their detailed consideration of this manuscript and for their highly supportive feedback. A point-by-point response to all reviewer comments is provided below.

Reviewer 1

1. *"In this paper titled "Nanoscale synchrotron x-ray analysis of intranuclear iron in melanised neurons of the Parkinson's substantia nigra", Jake Brooks et al. focuses on the accumulation of iron within the neurons of the substantia nigra in the human brain, which is a characteristic observed in Parkinson's disease (PD). The researchers used scanning transmission x-ray microscopy (STXM) to investigate the presence and nature of iron deposits in these neurons, providing sound evidence to identify the presence of multiple iron foci and their chemical properties for the first time. From the reviewer's perspective, this manuscript is suitable for publishing in Communications Biology with some minor revisions by answering the following questions."*

We thank the reviewer for their positive comments concerning the evidence presented in this manuscript.

2. *"In tissue preparation section, samples were all embedded in resin prior to ultramicrotome. Do you expect the iron oxidation state to remain intact during the embedding process? Have you considered the option to prepare and characterize the samples in frozen state?"*

We acknowledge that preserving the integrity of the iron chemistry is fundamental to this study. In prior work, we have performed extensive method development to ensure that the observations reported are robust and a feature of the system of interest, as opposed to an artefact of the resin embedding. The detail of this work was incorporated into our first publication concerning STXM analysis of iron in mammalian tissues (Telling et al., Cell Chemical Biology, 2017). We demonstrated that iron standards prepared using the same embedding series show no alteration in oxidation state when examined using STXM. On this basis, we expect the oxidation state to remain intact.

Analysis of frozen-hydrated tissue sections represents a significant technical advancement for STXM, and is not currently available at Diamond I08, where data for this study were collected. Logistical challenges associated with human brain tissue samples constitute an additional barrier. However, exciting ongoing development of cryogenic measurement capabilities at many synchrotron beamlines (see e.g. Leontowich et al. Review of Scientific Instruments, 2018) will soon create opportunities for the examination of vitrified biological specimens. In addition to the preservation of native sample (bio)chemistry and ultrastructure during sample preparation, performing STXM under cryogenic conditions also protects against X-ray damage induced by sample measurement at high X-ray doses,

allowing for increased tolerances to X-ray radiation (Hitchcock, et al., *Microscopy and Microanalysis*, 2020), which will improve data quality. We are currently working towards cryogenic-analysis in partnership with synchrotron beamline staff and hope to be in a position to trial STXM measurements under cryogenic conditions in the near future. We discuss the implementation of cryogenic STXM measurements and the benefits thereof, in our recent papers: Everett et al., *JESRP*, 2023 and Everett et al. *ACS Chemical Neuroscience*, 2024).

3. *"In STXM setup, could you explain how you can use a 50um probe to achieve 200nm spatial resolution maps? Additionally, could you also provide the dwell time, estimated dosage and energy resolution?"*

We thank the reviewer for highlighting this error in the text. The diameter of the focussed beam is approximately 50 nm, not 50 μm . This has been corrected in the revised manuscript (line 263).

The dwell time for all STXM maps was fixed at 10 ms.

For each presented x-ray absorption spectrum, energy resolution is typically not fixed across the entire spectrum. Energy resolution is increased (to a maximum achievable resolution of 0.1 eV) over the features of interest to resolve accurately the shape and position of the peaks, and then reduced (to a minimum of 1 eV) in the pre- and post-edge ranges. This makes it possible to acquire sufficient detail to interpret metal ion speciation from x-ray absorption spectra, whilst also permitting efficient use of limited synchrotron time. Full stack parameters are provided for each spectrum below. These stack parameters will also be accessible via the accompanying dataset upon publication.

Figure 1(f): 700-704 eV in 1 eV steps; 704.5-706 eV in 0.5 eV steps; 706.2-713 eV in 0.2 eV steps, 713.25-715.5 eV in 0.25 eV steps; 716-719 eV in 0.5 eV steps, 719.25-726 eV in 0.25 eV steps, 726.5-729 eV in 0.5 eV steps, 730-740 in 1 eV steps.

Figure 2:

Fe(0) spectrum: Fixed at 0.1 eV steps for entire spectrum.

Fe²⁺ spectrum: 700.5-703.15 eV in 0.5 eV steps; 703.35-713.1 eV in 0.15 eV steps; 713.45-718.25 eV in 0.4 eV steps; 718.35-735.15 eV in 0.15 eV steps.

Fe³⁺ spectrum: 700.65-703.65 eV in 1 eV steps; 704.75-713.55 eV in 0.1 eV steps; 714.15-719.15 eV in 0.5 eV steps; 719.95-733.65 eV in 0.3 eV steps.

Fe₃O₄ spectrum: Fixed at 0.1 eV for entire spectrum

Figure 3 (f, g): 700-705 eV in 1 eV steps; 705.5-707 eV in 0.5 eV steps; 707.125-713 eV in 0.125 eV steps; 713.25-715.5 eV in 0.25 eV steps; 716-719 eV in 0.5 eV steps; 719.25-726 eV in 0.25 eV steps; 726.5-729 eV in 0.5 eV steps; 730-740 eV in 1 eV steps.

Figure 4 (d, c): 700-704 eV in 1 eV steps; 704.5-706 eV in 0.5 eV steps, 706.2-713 eV in 0.2 eV steps, 713.25-715.5 eV in 0.25 eV steps, 716-719 eV in 0.5 eV steps, 719.25-726 eV in 0.25 eV steps, 726.5-729 eV in 0.5 eV steps, 730-740 in 1 eV steps.

Figure 5 (spectra 1 – 3): 700-705 eV in 1 eV steps; 705.5-714 eV in 0.5 eV steps; 715-718 in 1 eV steps.

We acknowledge the reviewer's request for estimated dosage, and understand that x-ray doses during sample measurement must be carefully controlled to ensure these doses do not significantly alter the chemistry of the material being examined.

However, in practice, it is extremely difficult to achieve any meaningful estimate of x-ray dosage due to intrinsic variability in the estimation of parameters that contribute to dose calculation, and the substantial variability in x-ray dose damage thresholds for differing elements and differing specimen preparations. X-ray imaging dose (D) can be estimated from the total interaction cross-section as follows:

$$D = \frac{I_0 t E}{A} \left[\frac{\mu}{\rho} \right]$$

Where I_0 is the incident photon intensity, E is the photon energy, $\left[\frac{\mu}{\rho} \right]$ is the x-ray mass attenuation coefficient, A is the sample area, and t is the exposure time. As described in Jones et al. Analytical Chemistry, 2017, whilst this dose estimate considers all of the modalities of energy transfer into the specimen, it makes no reference to the particulars of the interaction between the x-ray probe and the specimen: there is no distinction between dose that is deposited as heat, that which results in core or valence ionisation, or that which results in targeted bond-breaking within a system. This estimated dose calculation, carries at best an uncertainty order of 50%, where a specimen composition is unequivocally known. Even in homogenous specimens, small changes in the assumption of empirical formula can result in significant effects on dose estimates.

Here, we examined a complex sample of biological origin where the complete composition and structure cannot be fully determined; it is therefore impossible to provide an accurate dose estimate. Furthermore, our STXM measurements showed sample composition to be entirely heterogeneous, displaying variation at a nanoscale (i.e. pixel by pixel) level. Therefore, an example of a $5 \mu\text{m}^2$ scanning area, measured at 50 nm spatial resolution, would potentially provide 10,000 different estimates of x-ray dose. Even making the most

basic assumption that these samples were composed of two constituents: resin and tissue (of empirically-determined composition), the relative contribution of each of these constituents to the x-ray mass attenuation coefficient could vary from pixel to pixel.

By extension, if we were to provide a coarse x-ray dose estimate per scan (assuming uniform dose for each pixel of the scanning area), the heterogeneous nature of our sample material would mean the effect of this x-ray dose would also vary pixel by pixel.

Therefore, suitable x-ray dose(s) are empirically-tailored for each individual experiment dependent on the needs of the specific study and scientific questions under investigation.

We have an established track record in diligently assessing x-ray dose effects and controlling scanning parameters to match the needs of our experiments. Examples of this seen in the following:

Everett et al. *Science Advances*. 7,eabf6707(2021) Supplementary Materials text and Figures S11-S13; Brooks et al. *Angewandte Chemie*. 59, 11984-11991 (2020) Figure S7; Everett et al. *Scientific Reports*. 10, 10332 (2020) Figure S3.

We further demonstrate the successful optimisation of our scanning parameters (dosage) in Figure 5 of the current manuscript, where we demonstrate that the oxidation state of iron deposits within the sample material is maintained over three successive scans of the same area; demonstrating that the applied x-ray dosages were not sufficient to induce photoreduction of the iron.

4. *"In Fig 3 e&f, could you provide another baseline spectra on adjacent dark region of the iron map"*

This has now been added to Figure 3 e&f as requested (line 147 and shown below). The spectrum from the adjacent, extracellular region shows no detectable iron signal.

Figure 1: STXM characterisation of melanised neuron in Parkinson's SNc. **(a)** Protein map, showing cell membrane (yellow arrow), nucleus membrane (blue arrow) and nucleolus (orange arrow), **(b)** off-peak image at 530 eV showing distribution of neuromelanin, **(c)** iron map, **(d)** composite map showing protein (blue), neuromelanin (white), iron (red), **(e)** image of melanised cellular region highlighted in (d) at Fe^{3+} peak, **(f)** iron L_3 -edge spectra from neuromelanin clusters and extracellular region highlighted in (e). The dashed line at 708 eV and dotted line at 709.5 eV mark the principal absorption energies for Fe^{2+} and Fe^{3+} cations, respectively. **(g)** Iron $L_{2,3}$ -edge spectrum from intranuclear iron deposit highlighted in (c).

5. "Do you have an explanation why neuromelanin region shows iron traces in Fig 3 but not in Fig. 2?"

In this example, the iron signal associated with the neuromelanin clusters is faint and was thresholded out of the original Figure 1. In the revised manuscript (line 111), the lower

threshold on the iron map in Figure 1d has now been adjusted to capture the diffuse iron signal – the composite image in Figure 1e has been updated accordingly. The updated figure is shown below.

Whilst neuromelanin has a widely recognised high affinity for iron, it should be noted that not all melanised neurons harbour an equivalent iron load. The range in neuronal iron load in substantia nigra has also been shown to be particularly pronounced in the Parkinson's state (Oakley et al., *Neurology*, 2007). Hence the variation in iron signal amongst neurons is not unexpected.

Figure 2: Correlative mapping of cell nucleoli in adjacent 500 nm sections of Parkinson's SNc. **(a)** Melanised neuron stained for RNA, **(b)** protein map of same cell shown in (a) in adjacent unstained tissue section, blue and orange arrows mark the nuclear membrane and nucleolus, respectively, **(c)** off-peak image at 530 eV shows neuromelanin distribution and the cell nucleolus due to their relatively high optical density, **(d)** iron map, where the red arrow marks the position of intranuclear iron deposit, yellow asterisks mark the positions of iron containing blood vessels, **(e)** composite map showing protein (blue), density-related contrast (white) and iron (red), **(f)** iron L_{2,3}-edge spectrum from intranuclear iron deposit shown in (e), demonstrating the presence of ferric iron (see reference spectra in Figure 2).

Reviewer 2

6. *"The manuscript by Brooks et. al., describes a novel microscopy approach (STXM) to study sub-cellular Fe distribution in neuromelanin containing neurons in a case of Parkinson's disease. The novelty of the methods used should not be understated, and the data is of high value, revealing new information that is difficult to obtain by other methods."*

We thank the reviewer for their positive feedback concerning the novelty of both the approach and value of the findings presented in this manuscript.

7. *"Weaknesses of the paper are that a greater number of replicates were not analysed, and control neurons from healthy individuals were also not analysed. The reviewer acknowledges that obtaining access to such samples, and sufficient access to the analytical equipment is prohibitive though"*

We appreciate that we had not included data from controls in this manuscript, and as such made no claims that findings of intranuclear iron made in this study are disease-specific. Nevertheless, we feel the topic is important in the context of better understanding the role of iron in both disease and in normal brain function and the findings presented along with the discussion material will stimulate further investigation (in full agreement with the reviewer's subsequent comment below).

We would also like to take this opportunity to present STXM data from two control neurologically-healthy control cases (see Figure 1 below), with intranuclear iron not observed in either case. However, we deliberately chose not to include these data in the manuscript as they imply a comparison between disease and control state that we cannot legitimately make with the available dataset. In addition to the small sample size, tissue sections were cut to sub-micron thickness (as required for STXM). Small iron deposits may therefore be present in the control samples but out of the plane of the tissue section. It is not anticipated that the advanced x-ray methods being applied here can be used for high-throughput studies (i.e. to enable group sizes >10), therefore direct comparison of intranuclear iron content in disease versus control state for a statistically significant number of neurons is currently unrealistic. This would require technical developments that might soon become feasible (as demonstrated by delivery of high-throughput synchrotron measurements for protein crystallography) but are beyond the scope of this particular study.

As supported by the reviewer comment below, the value in this approach is instead in determining the composition, speciation, and distribution of metal-rich deposits at the organelle level, whilst also capturing the anatomical context without requirement for staining, ablation, or for correlative analysis. This cannot be achieved with any other

technique and this manuscript serves to showcase that unique capability and applicability to unanswered questions in biology.

Figure 1: STXM characterisation of melanised neurons in neurologically-healthy control SNc: Case C1 **(a-b)** Case C2 **(c-d)**. **(a)** Protein map, nucleolus marked by an orange arrow **(b)** iron map of full area shown in (a), **(c)** single image at 705 eV, nucleolus marked by an orange arrow, **(d)** iron map of area shown in (c).

8. *“Although only a limited number of cells were imaged, the association of Fe with the nucleolus and RNA is very interesting. As mentioned above, the analytical capability to directly resolve sub-cellular Fe distribution, and to visualise its accumulation next to the nucleolus is very novel. I feel the identification of the Fe enrichment next to the nucleolus is a significant finding, and will enable other studies to further investigate this topic.”*

We thank the reviewer for recognising the value in showcasing the analytical capability and fully agree that it will enable future studies to further investigate this important topic.

9. *"I am however, surprised that the manuscript does not contain a discussion on iron responsive elements (IREs) / iron responsive proteins (IRPs) and how this may relate to nucleolar Fe content – I think the discussion would benefit if a paragraph was added on this topic."*

We thank the reviewer for their insightful suggestion to include this additional material and agree that it enriches the discussion.

As such, we have added the text below to the amended version of the manuscript (lines 201 – 206):

"Interaction between iron regulatory proteins (IRP1 and IRP2) and iron responsive elements (IREs) present in the untranslated region of certain mRNAs is central to intracellular iron homeostasis [1, 2]. IRPs have RNA-binding properties that depend on the presence of a 4Fe-4S cluster [1]. Whilst IRPs are typically regarded as cytosolic proteins, nuclear localisation of IRP1 has been observed in iron-replete cells, suggesting a cell-specific response mediated by an iron-dependent mechanism [1]."

[1] Gu, W., C. Fillebeen, and K. Pantopoulos, Human IRP1 translocates to the nucleus in a cell-specific and iron-dependent manner. *International Journal of Molecular Sciences*, 2022. 23(18): p. 10740.

[2] Pantopoulos, K., Iron metabolism and the IRE/IRP regulatory system: an update. *Annals of the New York Academy of Sciences*, 2004. 1012(1): p. 1-13.

10. *"The following statement on page 4 of the PDF should have a reference. "Whilst use of stains or chemical fixatives is known to alter significantly native tissue chemistry, synchrotron techniques can be applied without requirement for staining or sample ablation"*

We agree with the reviewer and have added the following reference to support this section of text (line 74).

*"Chwiej, J., Szczerbowska-Boruchowska, M., Lankosz, M., Wojcik, S., Falkenberg, G., Stegowski, Z. and Setkowicz, Z., 2005. Preparation of tissue samples for X-ray fluorescence microscopy. *Spectrochimica Acta Part B: Atomic Spectroscopy*, 60(12), pp.1531-1537."*

11. *"Presumably a large portion of the Fe in the blood vessel in Figure 5 is heme-bound Fe. Are there any spectral features in Figure 5 that help confirm the speciation of Fe as heme (e.g., hemoglobin)."*

Prior work by Hocking and colleagues (Hocking et al., Journal of the American Chemical Society, 2007) compares iron L-edge spectra for heme and non-heme iron compounds. Findings demonstrate that the iron L-edge spectrum from the example heme iron compound [Fe(tpp)(ImH)₂] decreases in total intensity and shifts by 0.1 eV to a lower energy compared to the non-heme reference.

This very subtle energy shift is lower than the minimum energy step used to collect spectra presented in Figure 5 of this manuscript. Therefore, whilst theoretically possible to distinguish between heme and non-heme sources using iron L-edge spectroscopy, this could not be concluded from the available dataset. We emphasise that the spectra presented in Figure 5 are used solely to validate the measurement parameters, and do not impact upon the conclusions drawn from the study.

12. *"Lastly, the overall "tone" of the manuscript is that Fe accumulation is "bad", and that the Fe enrichment near the nucleolus is a component of PD pathology. However, as the authors haven't studied healthy neurons this remains to be shown. Further, although Fe accumulation can be "bad", Fe is essential for cell function, especially neuronal function. I feel like the discussion could be better balanced to acknowledge that the Fe accumulation near the nucleolus might not be a consequence of pathology and could indeed be involved in healthy function of the neuron."*

We thank the reviewer for bringing this to our attention. In full understanding that iron enrichment could be constructive or destructive, we made considerable effort to balance the argument in the Discussion section of the manuscript.

Following the statement in the Discussion *"the potential for iron to play a constructive, physiological role inside the nucleus cannot be excluded"*, we have described several roles by which intranuclear iron could, as the reviewer correctly points out, be involved in healthy neuronal function. These include, as a repository for iron-dependent DNA synthesis proteins, to protect DNA from oxidative damage oxidation, and potential involvement in RNA biosynthesis.

With the addition of the paragraph above concerning possible intranuclear localisation of IRP (as suggested by the reviewer), we hope that the tone of the discussion section is adequately balanced.